# BET Inhibitor JQ1 Attenuates Feline Leukemia Virus DNA, Provirus, and Antigen Production in Domestic Cat Cell Lines

**DOI:** 10.3390/v15091853

**Published:** 2023-08-31

**Authors:** Garrick M. Moll, Cheryl L. Swenson, Vilma Yuzbasiyan-Gurkan

**Affiliations:** 1Comparative Medicine & Integrative Biology, College of Veterinary Medicine, Michigan State University, East Lansing, MI 48824, USA; mollgarr@msu.edu; 2Department of Pathobiology & Diagnostic Investigation, College of Veterinary Medicine, Michigan State University, East Lansing, MI 48824, USA; 3Veterinary Diagnostic Laboratory, College of Veterinary Medicine, Michigan State University, East Lansing, MI 48824, USA; 4Department of Small Animal Clinical Sciences, College of Veterinary Medicine, Michigan State University, East Lansing, MI 48824, USA; 5Department of Microbiology & Molecular Genetics, Michigan State University, East Lansing, MI 48824, USA

**Keywords:** feline leukemia virus, retrovirus, gammaretroviruses, provirus, viral load, integration, epigenetics, bromodomains, BET inhibitors

## Abstract

Feline leukemia virus (FeLV) is a cosmopolitan gammaretrovirus that causes lifelong infections and fatal diseases, including leukemias, lymphomas, immunodeficiencies, and anemias, in domestic and wild felids. There is currently no definitive treatment for FeLV, and while existing vaccines reduce the prevalence of progressive infections, they neither provide sterilizing immunity nor prevent regressive infections that result in viral reservoirs with the potential for reactivation, transmission, and the development of associated clinical diseases. Previous studies of murine leukemia virus (MuLV) established that host cell epigenetic reader bromodomain and extra-terminal domain (BET) proteins facilitate MuLV replication by promoting proviral integration. Here, we provide evidence that this facilitatory effect of BET proteins extends to FeLV. Treatment with the archetypal BET protein bromodomain inhibitor (+)-JQ1 and FeLV challenge of two phenotypically disparate feline cell lines, 81C fibroblasts and 3201 lymphoma cells, significantly reduced FeLV proviral load, total FeLV DNA load, and p27 capsid protein expression at nonlethal concentrations. Moreover, significant decreases in FeLV proviral integration were documented in 81C and 3201 cells. These findings elucidate the importance of BET proteins for efficient FeLV replication, including proviral integration, and provide a potential target for treating FeLV infections.

## 1. Introduction

Feline leukemia virus (FeLV) is a cosmopolitan pathogenic gammaretrovirus with the potential to establish persistent lifelong infections in domestic and wild felids (including endangered species), causing fatal illnesses and a predisposition to morbidities via immunosuppression. Approximately 33% of domestic cats exposed to FeLV become persistently viremic, with subsequent development of fatal FeLV-associated diseases, including immunodeficiencies, anemias, leukemias, and lymphomas. The median survival time for over 800 FeLV-infected cats was reportedly 2.4 years, compared to 6.3 years for more than 7000 sex- and age-matched uninfected control cats in a survey conducted in the United States [1]. There is currently no definitive treatment for FeLV, and while existing vaccines reduce the prevalence of progressive infections, they neither provide sterilizing immunity nor prevent regressive infections; this results in viral reservoirs with the potential for reactivation, transmission, and disease. Therefore, studies aimed at treatment(s) to reduce the FeLV load of infected cats are essential for reducing FeLV-associated diseases and fatalities.

Approximately two-thirds of domestic cats exposed to FeLV develop abortive or regressive infections by mounting rapid, FeLV-directed, cytotoxic T lymphocyte and neutralizing antibody immune responses that minimize infection [2]. By contrast, approximately one-third of domestic cats fail to generate sufficient cell-mediated and humoral immune responses to curb FeLV proliferation following exposure; these cats experience progressive infection with 100% morbidity and mortality [2,3].

Cats with progressive FeLV infections have persistently high viral DNA loads and viral titers, whereas those with regressive infections have low viral DNA loads that reportedly correlate with low viral titers [3,4,5,6,7,8]. Notably, total FeLV DNA load and FeLV p27 capsid concentration are highly correlated with survival times in naturally infected cats [9]. Moreover, cats with regressive FeLV infections may develop malignancies if proviral integration occurs proximal to or within host proto-oncogenes due to *cis*-acting proviral elements (i.e., insertional mutagenesis). Indeed, cats that are not progressively FeLV-infected but live with progressively infected cats have over a 40-fold increased risk of lymphoma compared to those not exposed to FeLV [10].

Previous studies suggest that a prolonged reduction in viral load may enable conversion from a progressive to a regressive FeLV infection in some cats [11,12]. Integrase inhibitors such as raltegravir, used to treat HIV-1 in humans, decrease HIV-1 viral RNA and proviral loads [13,14]. Short-term raltegravir treatment of progressively FeLV-infected cats decreased FeLV RNA loads. Following cessation of treatment, viral RNA loads returned to pretreatment levels in seven cats, while one cat developed antibodies to FeLV and maintained a reduced viral RNA load [11]. This suggests that integration inhibitors may attenuate FeLV proviral load and virus production in progressively infected cats.

Earlier reports established that murine leukemia virus (MuLV) and FeLV retroviral integrases interact with and bind to the highly conserved extra-terminal (ET) domain of epigenetic reader bromodomain and extra-terminal domain (BET) proteins BRD2, BRD3, and BRD4 (~90% identity among mammalian species; see Appendix A) [15,16,17,18,19,20]. Moreover, this binding interaction enhances the efficiency of MuLV proviral integration into the host genome by tethering MuLV pre-integration complexes to chromatin via the bromodomains and allosterically activating retroviral integrase via the ET domain [16].

BET protein bromodomain inhibitors (BETi) are a class of small molecules that selectively disrupt BET protein bromodomains binding to N-ε-acetyl-L-lysine protein residues, including those on histones, thereby displacing BET proteins from chromatin [21]. Notably, significant dose-dependent decreases in proviral load and virus expression were reported in MuLV-exposed cells treated with nanomolar concentrations of BETi compared to vehicle controls [15,16,17]. However, the roles of BET proteins and the effects of BETi on FeLV infection have not previously been reported. Here we present data supporting that BETi (+)-JQ1 treatment decreases FeLV proviral load, total FeLV DNA load, and p27 capsid protein expression in two phenotypically divergent domestic cat cell lines challenged with infectious virus. Importantly, FeLV proviral integration was significantly decreased in (+)-JQ1-treated 81C and 3201 cells challenged with infectious FeLV, suggesting that this mechanism may play an important role in the observed reductions in total viral DNA, proviral DNA, and p27 antigen concentrations.

## 2. Materials and Methods

### 2.1. Cell Culture and Virus

The FeLV-negative feline kidney fibroblast cell line 81C (RRID:CVCL_7216) was derived as a subclone of the one-hit murine sarcoma virus-transformed 8C subline of the Crandell-Rees Feline Kidney cell line that originated from the kidney tissue of a healthy young domestic cat. The cell line proliferates as adherent cells in medium, as previously described [22,23].

The FeLV-negative lymphoma cell line 3201 (RRID:CVCL_X612) was derived from a naturally acquired domestic feline thymic lymphoma and propagated as suspended cells in medium as previously reported, with McCoy’s 5A substituted for RPMI 1640 [24].

Persistently FeLV-infected feline lymphoma cell line FL-74 (RRID:CVCL_Y090), originally derived from an experimentally FeLV-induced lymphoma, produces subgroups A, B, and C infectious FeLV [25,26]. The cell line grows as suspended cells in culture medium composed of 87.9% (*v*/*v*) RPMI 1640 (Gibco, Thermo Fisher Scientific, Waltham, MA, USA), 10% (*v*/*v*) heat-inactivated fetal bovine serum (New Zealand- or Australia-sourced; Gibco, Thermo Fisher Scientific, Waltham, MA, USA), and final concentrations of 3.81 mM L-glutamine, 50 μg/mL gentamicin, 100 IU/mL penicillin, 100 μg/mL streptomycin, and 0.25 μg/mL amphotericin B. Cell-free FL-74 cell-conditioned medium was used for the FeLV virus challenge in all experiments.

Cell lines were generously provided by Dr. Lawrence E. Mathes at the Ohio State University. Cell lines were tested for *Mycoplasma* spp. contamination by DAPI staining prior to each experiment. All cell cultures were maintained at 37 °C and 5% (*v*/*v*) CO_2_.

### 2.2. Cell Counts

Viable adherent cell concentrations (81C cells) and total and viable suspended cell concentrations (3201 cells) were quantified in technical duplicate for each treatment and timepoint by trypan blue exclusion using a Countess automated cell counter (Invitrogen, Thermo Fisher Scientific, Waltham, MA, USA). Nonadherent 81C cells were discarded.

### 2.3. Treatments

Pan-BETi (+)-JQ1 and its biologically inactive enantiomer (−)-JQ1 (Cayman Chemical, Ann Arbor, MI, USA) were utilized at 31.25 nM, 125 nM, or 500 nM final concentrations; the final vehicle concentration was 0.005% (*v*/*v*) DMSO. Negative and positive controls were untreated DMSO-negative virus-negative and untreated DMSO-negative virus-positive, respectively. Additional untreated DMSO-positive virus-negative and untreated DMSO-positive virus-positive groups were utilized for 3201 cell experiments.

### 2.4. Creation of Virus Stocks

FL-74 cells were passaged to a starting cell concentration of 2.1 × 10^6^ ± 5% viable cells/mL in 30 mL culture medium in T175 flasks. Conditioned FL-74 cell medium was harvested 40–48 h after cell passage, filtered (0.45 μm polyvinylidene fluoride membrane), aliquoted, snap frozen, and stored at −80 °C until use as viral stock. Viral stocks were thawed immediately prior to the challenge.

### 2.5. Treatment and Virus Challenge of Cell Lines

81C cells (1.3 × 10^5^ viable cells/well) were seeded into 12-well culture plates to accommodate three biological replicate samples for each treatment at each collection timepoint. When cells reached approximately 75–80% confluency, cell culture medium was removed and replaced with 0.8 mL of fresh medium containing (+)-JQ1 or the enantiomer (−)-JQ1 at final concentrations of 0 (untreated virus-positive and virus-negative controls), 31.25, 125, or 500 nM, and 4 μg/mL hexadimethrine bromide, and incubated for 2 h. Except for virus-negative controls, cell cultures were then challenged with 150 µL FL-74 viral stock, 4 μg/mL hexadimethrine bromide, and respective concentrations of (+)-JQ1 or (−)-JQ1, with a final volume of 1 mL, and incubated for an additional 2 h. Wells were gently rinsed twice with corresponding compound-containing cell culture media, and incubation was continued using medium containing respective (+)-JQ1 or enantiomer concentrations. Half-volume (0.5 mL) fresh cell culture medium changes with appropriate (+)-JQ1 or (−)-JQ1 concentrations were performed at 72, 120, and 192 h post-infection (hpi). At each data timepoint, cell culture medium was collected, centrifuged (2 min, 8000 rcf), and the supernatant stored at −80 °C until analysis. Wells were washed with PBS, cells were incubated with 750 µL 0.05% (*w/v*) trypsin (Gibco, Thermo Fisher Scientific, Waltham, MA, USA) for 5 min, and trypsin was inactivated by adding 250 µL of complete culture medium containing 4× appropriate (+)-JQ1 or (−)-JQ1 concentrations. Resuspended viable cell concentrations were measured, and aliquots were centrifuged (2 min, 8000 rcf), supernatants decanted, and cell pellets stored at −80 °C until further analysis.

3201 cells (2.0 × 10^6^ viable cells/well) were seeded into 6-well culture plates in 2 mL cell culture medium containing 4 μg/mL hexadimethrine bromide and 0, 31.25, 125, and 500 nM (+)-JQ1, (−)-JQ1, or DMSO vehicle only, and incubated for 2 h. Except for virus-negative control wells, cell cultures were challenged with 571.5 µL FL-74 viral stock, 4 μg/mL hexadimethrine bromide in (+)-JQ1, (−)-JQ1, or DMSO vehicle, and incubated for an additional 2 h. Cells were washed and centrifuged (5 min, 150 rcf) three times in complete culture media containing the respective (+)-JQ1, enantiomer, or DMSO vehicle. Cells were then resuspended in 4 mL cell culture medium containing appropriate (+)-JQ1, (−)-JQ1, or DMSO vehicle concentrations, and incubation continued. At designated timepoints, 500 μL of well-mixed cell suspensions were collected to determine total and viable cell concentrations and harvest supernatants and cell pellets for storage at −80 °C until use. Experiments were performed in biological triplicate.

### 2.6. FeLV p27 Enzyme-Linked Immunosorbent Assay (ELISA)

A commercially available kit, the ViraCHEK FeLV Antigen Test Kit (Zoetis, Parsippany, NJ, USA), was used to measure FeLV p27 antigen concentrations according to directions with the following modifications: culture supernatant samples were added neat or diluted in complete cell medium to maintain measurements within the logarithmic range of the assay; well plates were agitated at 600 rpm for 3 min and then incubated for 2 h; chromogenic substrate buffer was diluted to 5% (*v*/*v*) using ultrapure water; and reactions were stopped at 5 min with 25 μL of 4N H_2_SO_4_. The optical density (OD) of each well was measured at 450 nm (reaction product) and 650 nm (background) with an Envision spectrophotometer (PerkinElmer, Waltham, MA, USA). A weighted four-parameter logistic standard OD versus FeLV p27 concentration curve was created using serial dilutions of recombinant FeLV p27 (Fitzgerald Industries International, Acton, MA, USA) in complete cell medium and utilized to calculate sample concentrations from measured ODs.

### 2.7. Normalization of FeLV p27 Concentration

FeLV p27 antigen concentrations were normalized to the corresponding viable adherent (81C) or viable suspended (3201) cell number for each treatment concentration and timepoint.

### 2.8. DNA Extraction and Measurement

DNA was extracted using the DNeasy Blood & Tissue Kit (Qiagen, Hilden, Germany) and quantified (Qubit dsDNA HS Assay Kit, Qubit 2.0; Thermo Fisher Scientific, Waltham, MA, USA) per manufacturer protocols.

### 2.9. Creation of Plasmid Standards for Absolute Quantification and Normalization of Total FeLV DNA

Extracted DNA from FeLV+ domestic cat T cells was used to create PCR amplicons consisting of a 468-bp FeLV long terminal repeat (LTR) sequence containing an exogenous (hereinafter denoted as FeLV DNA) FeLV-specific 131-bp U3 region as described and adapted for template sequences [27,28]. The single-copy per haploid genome domestic feline albumin (fALB) gene was used for absolute quantification of feline DNA equivalents; 150 bp fALB amplicons were amplified from feline DNA using fALB-specific primers [29].

FeLV LTR or fALB amplicons were ligated into the pCR4-TOPO cloning vector (Invitrogen, Thermo Fisher Scientific, Waltham, MA, USA) and transfected into chemically competent cells (Endura, LGC Biosearch Technologies, Teddington, MDX, UK), which were then plated on kanamycin plates and incubated overnight at 37 °C; positive colonies were selected by kanamycin resistance and *ccdB* gene disruption. Sequences of cloned DNA amplicons were verified by Sanger sequencing (Michigan State University, Research Technology Support Facility, Genomics Core, East Lansing, MI, USA). A single transformant colony was used to inoculate a starter culture (kanamycin-LB medium; 8 h incubation; 37 °C; 300 rpm) that was then diluted 1:500 in kanamycin-LB medium and incubated (16 h; 37 °C; 300 rpm); the resultant plasmids were extracted (Plasmid Midi Kit, Qiagen, Hilden, Germany), linearized with *NotI*-HF (New England Biolabs, Ipswich, MA, USA), and agarose gel purified (QIAquick Gel Extraction Kit, Qiagen, Hilden, Germany).

Copy numbers were calculated using a NanoDrop spectrophotometer, Qubit 2.0 fluorometer (Invitrogen, Thermo Scientific, Waltham, MA, USA), and agarose gel electrophoresis (Typhoon FLA 9500, Cytiva, Marlborough, MA, USA). Ten-fold serial dilutions of FeLV LTR and fALB standard DNA templates were prepared in 30 μg/mL chum salmon (*Oncerhyncus keta*) sperm DNA, aliquoted, and stored at −20 °C until use.

### 2.10. Quantitative PCR (qPCR) for Absolute Quantification and Normalization of Total FeLV DNA Load

Total FeLV copy number for each sample was quantified using a modified singleplex qPCR FAM-TAMRA fluorogenic hydrolysis probe system adapted for template sequences (Table A1) [27]; a separate singleplex qPCR VIC-TAMRA fluorogenic hydrolysis probe system determined fALB gene copy number (Table A1) [29]. Accounting for two FeLV LTR U3 copies per FeLV DNA genome and two fALB copies per domestic cat cell, the normalized total FeLV DNA load per cell was calculated as: $\frac{FeLV\;LTR\;U3\;Copy\;Number\;}{fALB\;Copy\;Number}$.

qPCR assays consisted of respective primers, a fluorogenic hydrolysis probe (FAM- or VIC-TAMRA), 1× TaqMan Universal PCR Master Mix (Applied Biosystems, Thermo Fisher Scientific, Waltham, MA, USA), and 20 ng input DNA.

Standard curve and sample qPCR assays were performed in technical duplicate on either a QuantStudio 3 or QuantStudio 7 Real-Time PCR System (Applied Biosystems, Thermo Fisher Scientific, Waltham, MA, USA).

### 2.11. Relative Quantification of FeLV Proviral Load

To determine the effects of BETi on FeLV proviral integration, technical duplicate PCR reactions were performed for each biological triplicate using primer sequences adapted for the template as necessary (Table A2) [4]. Briefly, FeLV proviral DNA was amplified in a preamplification reaction (‘SINE’) using a reverse primer specific for domestic feline short interspersed nuclear elements (SINE), and a FeLV LTR U3 region-specific forward primer 5′-appended with a λ-bacteriophage sequence. PCR products synthesized from unintegrated FeLV DNA were accounted for by using a parallel reaction (‘NOISE’) that substituted the SINE-specific forward primer with a λ-bacteriophage sequence non-complementary to feline genomic DNA. Quantitation of preamplification reaction products was performed by qPCR using the U3-specific λ-bacteriophage-appended forward primer with the reverse ‘SINE’ primer and a FeLV LTR U3 region-specific fluorogenic hydrolysis probe.

The difference between $C_{T}$ values of parallel ‘SINE’ and ‘NOISE’ reactions reflects the relative abundance of proviral to total FeLV DNA ($\Delta C_{T_{R}}$), and by extension, can indirectly reflect the relative abundance of proviral to unintegrated FeLV DNA.

‘SINE‘ and ‘NOISE’ preamplification reactions contained primers (SINE1-Fo, SINE1-Re; or NOISE-Fo, NOISE-Re—200 μM each; Table A2), 1.5 mM MgCl_2_, 250 μM dNTPs, 1× PCR Buffer, 0.125 U/μL native *Taq* Polymerase (Invitrogen, Thermo Fisher Scientific, Waltham, MA, USA), and 20 ng input DNA. Reactions were performed in technical duplicate (SimpliAmp or Veriti Thermal Cycler, Applied Biosystems, Thermo Fisher Scientific, Waltham, MA, USA).

Nested qPCRs contained primers (IntFeLV-Fo, IntFeLV-Re—480 nM each; Table A2), a FAM-TAMRA fluorogenic hydrolysis probe (FeLVU3-Pr—160 nM), 1× TaqMan Universal PCR Master Mix, and 2 μL of preamplicons from the above preamplifications. Reactions were performed in technical duplicate (QuantStudio 3 or QuantStudio 7 Real-Time PCR System, Applied Biosystems, Thermo Fisher Scientific, Waltham, MA, USA).

The $2^{{-∆∆C}_{T}}$ method was used to analyze and relatively quantify FeLV proviral DNA (‘SINE’) and unintegrated FeLV DNA [30]. Data for each experiment were calibrated to the mean untreated FeLV-exposed control at the earliest measured timepoint (see figure legends). Individual $\Delta C_{T}$ values for ‘SINE’ were calculated as $\Delta C_{T_{SINE}}=C_{T_{SINE}}-C_{T_{fALB}}$, while $\Delta \Delta C_{T_{SINE}}$ values were calculated as $\Delta \Delta C_{T_{SINE}}={\Delta C}_{T_{SINE}}-{\Delta C}_{T_{CALIBRATOR}}$. Individual $\Delta C_{T_{R}}$ values were calculated as $\Delta C_{T_{R}}=C_{T_{NOISE}}-C_{T_{SINE}}$, while $\Delta \Delta C_{T_{R}}$ values were calculated as $\Delta \Delta C_{T_{R}}={\Delta C}_{T_{R}}-{\Delta C}_{T_{CALIBRATOR}}$.

### 2.12. Statistical Analyses

Data were analyzed with one-way ANOVA and Holm-Šídák post hoc analysis for significance of multiple comparisons, i.e., D(±)V(+)T(−) vs. [(+)-JQ1]; and [(−)-JQ1] vs. [(+)-JQ1] [31,32]. Significance was α = 0.05; reported statistical significances and *p*-values were determined by the greatest comparison *p*-value for each [(+)-JQ1]. Assumptions for statistical analyses were tested using the Brown–Forsythe test for variance and Shapiro–Wilk’s W test for normality [33,34]. Prism v9.5.1 (GraphPad Software, San Diego, CA, USA) was used for all statistical analyses. Stereoisomer (−)-JQ1 treatment data are displayed as a single group.

## 3. Results

### 3.1. Effects of (+)-JQ1 on 81C Cell Cultures Challenged with FeLV

#### 3.1.1. Effect on Viable Adherent Cell Counts

Gammaretroviral DNA gains access to integrate into host cell chromatin and become proviral DNA during nuclear membrane dissolution in the metaphase stage of cell division. Therefore, we measured the effects of (+)-JQ1 on viable cell numbers.

Viable (adherent) cell counts of 500 nM (+)-JQ1-treated 81C cell cultures were significantly lower than enantiomer (−)-JQ1-treated and untreated virus-negative and virus-positive untreated control cultures at 48 and 72 hpi (logarithmic growth phase), while no differences in viable cell concentrations were seen at 192 hpi (plateau phase) when all treated and virus-exposed untreated control cultures had expanded to form 100% confluent monolayers. Notably, no significant differences in viable 81C cell counts were observed for 31.25 or 125 nM (+)-JQ1 at any measured timepoint compared to enantiomer and untreated virus-exposed controls (Figure 1a). By contrast, viable (adherent) cell counts of untreated virus-unexposed cultures were greater than all virus-exposed cultures at 192 hpi.

#### 3.1.2. Effect on FeLV p27 Capsid Protein Expression

The FeLV p27 capsid protein concentration normalized for viable adherent 81C cell concentration was significantly lower than untreated and (−)-JQ1 enantiomer controls at 72 and 192 hpi in cultures treated with 500 nM (+)-JQ1, as well as at 192 hpi in cultures treated with 125 nM (+)-JQ1 (Figure 1b).

#### 3.1.3. Effect on Total FeLV DNA Load 

The effect of (+)-JQ1 on total FeLV DNA load was absolutely quantified and normalized to the single copy per haploid genome feline albumin (fALB) gene [29], in order to calculate total FeLV DNA load per cell. Significant dose-dependent decreases in total FeLV DNA load of (+)-JQ1-treated, FeLV-exposed 81C cells were observed at 24 hpi for 500 nM; 48, 72, and 192 hpi for 500 and 125 nM; and 192 hpi for 31.25 nM (+)-JQ1 compared to enantiomer and untreated controls (Figure 1c).

The inhibitory effects of (+)-JQ1 on normalized FeLV p27 antigen expression and total viral DNA load were generally greater than the suppressive effects on viable adherent 81C cell counts over time.

#### 3.1.4. Effect on Proviral Integration

In order to assess the effects of BETi (+)-JQ1 on FeLV integration, proviral (integrated) FeLV DNA was quantified using a nested SINE-LTR PCR system developed by Cattori et al. based on nested *Alu*-LTR PCR systems for quantification of proviral HIV-1 DNA [4,35].

Significant dose-dependent decreases in fALB-normalized FeLV proviral loads of 81C cell cultures treated with 125 nM and 500 nM (+)-JQ1 and challenged with FeLV were observed at 48, 72, and 192 hpi compared to enantiomer and untreated controls (Figure 2a). A dose-dependent increase in the relative total versus proviral FeLV DNA ratio (calibrated to the mean untreated FeLV-exposed control) at 24 hpi, an estimate of excess unintegrated FeLV DNA forms compared to controls, was observed at 125 nM (+)-JQ1 and reached significance at 500 nM (+)-JQ1 (Figure 2b). These data suggest that (+)-JQ1 impairs proviral integration in 81C cells.

Together, these data suggest that the (+)-JQ1-induced reduction of 81C cell proviral load, total FeLV DNA, and subsequent p27 capsid protein production may be due to inhibition of FeLV proviral integration.

### 3.2. Effects of (+)-JQ1 on 3201 Cell Cultures Challenged with FeLV

#### 3.2.1. Effects of DMSO Vehicle

No significant differences in viral infection were observed in two additional control groups, FeLV-exposed and FeLV-unexposed 3201 cells with DMSO vehicle alone at the concentration utilized in our experiments [36,37,38].

#### 3.2.2. Effect on Total and Viable Cell Counts

A significant dose-dependent decrease in total viable 3201 cell counts was observed at 72 hpi for 500 nM and at 96 hpi for 125 and 500 nM (+)-JQ1 compared to controls (Figure 3a); however, no significant differences in percent cell viability were detected at any measured (+)-JQ1 concentration or timepoint compared to controls (Appendix A).

Together, these data suggest an inhibitory effect of 500 and 125 nM (+)-JQ1 on 3201 cell proliferation over time.

#### 3.2.3. Effect on FeLV p27 Capsid Protein Expression

FeLV p27 capsid protein expression normalized for viable cell count for 500 nM (+)-JQ1-treated cells was significantly lower at 48, 72, and 96 hpi (Figure 3b) compared to untreated, enantiomer, or vehicle controls.

#### 3.2.4. Effect on Total FeLV DNA Load

Total fALB-normalized FeLV DNA loads of FeLV-exposed 3201 cells were significantly decreased at 72 and 96 hpi for cell cultures treated with 500 nM (+)-JQ1 compared to controls. Significant increases in FeLV DNA load were seen at 24 hpi for 500 nM (+)-JQ1 and at 96 hpi for 31.25 nM (+)-JQ1 (Figure 3c).

The inhibitory effects of (+)-JQ1 on normalized FeLV p27 capsid protein expression and total viral DNA load generally exceeded the suppression of 3201 cell proliferation.

#### 3.2.5. Effect on Proviral Integration

Although not statistically significant, the fALB-normalized FeLV proviral load in 500 nM (+)-JQ1-treated, FeLV-exposed 3201 cells trended lower compared to controls at all measured timepoints (Figure 4a). The proportion of unintegrated FeLV DNA in 3201 cells treated with 500 nM (+)-JQ1 was significantly greater at 24 hpi compared to controls (Figure 4b). Together, these data suggest that (+)-JQ1 also impairs proviral integration in 3201 cells.

Collectively, these data suggest that (+)-JQ1 inhibits proviral integration in 3201 cells, thereby causing reductions in proviral load, total FeLV DNA, and FeLV p27 capsid protein expression.

## 4. Discussion

Normalized FeLV p27 capsid protein expression and total FeLV DNA load of adherent fibroblast (81C) and suspended lymphoma (3201) cells treated with the BETi (+)-JQ1 and challenged with FeLV were attenuated in a dose-dependent manner that exceeded antiproliferative effects. Significant dose-dependent decreases in normalized FeLV proviral loads in (+)-JQ1-treated 81C cells exposed to FeLV were observed, while equivalently treated 3201 cells showed a trend toward reduced proviral loads. Notably, FeLV infection of 81C cells was attenuated with 125 nM (+)-JQ1 treatment without affecting viable adherent cell counts (Figure 1c), indicating that the mechanism of FeLV attenuation is not dependent on alterations in cell proliferation. While normalized total FeLV DNA was decreased in 3201 cell cultures treated with 500 nM (+)-JQ1 and challenged with FeLV by 72 and 96 hpi, increases were seen in 31.25 nM (+)-JQ1-treated 3201 cell cultures at the earliest measured timepoint (24 hpi) and at 96 hpi (Figure 3c); this may be in part due to a more rapid replication rate of 3201 compared to 81C cells, experimental variability, or related to the observed greater proportion of unintegrated versus proviral FeLV DNA.

Our data suggest that (+)-JQ1 may act by impairing FeLV proviral integration into the host cell genome via BET protein bromodomain inhibition, thereby causing decreased total FeLV DNA load, p27 antigen expression, and possibly FeLV proviral load, similar to MuLV [17]. This is the first report indicating a potential facilitative role for BET proteins in FeLV infection.

Normalized total FeLV DNA of 81C cells treated with 125 nM (+)-JQ1 and challenged with FeLV was not significantly different compared to controls at 24 hpi; however, normalized total FeLV DNA, p27 antigen, and proviral loads were reduced at subsequent timepoints (Figure 1c). These data suggest that viral entry, uncoating, and reverse transcription are neither affected by nor involved in the attenuative effects of (+)-JQ1 on FeLV infection. This is consistent with previous MuLV experiments in which the amounts of early timepoint total MuLV DNA and early reverse transcription products, i.e., minus-strand strong-stop extension (MSSE) and plus-strand extension (PSE) products, were unaffected by (+)-JQ1 treatment [15,16,17]. Notably, the relative proportion of unintegrated FeLV DNA was significantly increased compared to controls in both 81C and 3201 cells treated with 500 nM (+)-JQ1 (Figure 2b and Figure 4b), suggesting impaired proviral integration. This is concordant with a previous study on MuLV that showed (+)-JQ1 treatment resulted in significantly increased 2-LTR circles, markers indicative of defective proviral integration [17]. Together, these findings suggest that (+)-JQ1 inhibition of BET proteins suppresses FeLV replication at the proviral integration step, as reported for MuLV.

The molecular interaction of the extra-terminal (ET) domains of BRD2 and BRD4 with retroviral integrase was previously described as limited to and conserved among gammaretroviruses, including FeLV. In fact, site-directed mutagenesis mapping of the interface for murine BRD2 binding with MuLV and FeLV integrases suggests that gammaretroviral integrases share a conserved binding site in the ET domain [16]. Moreover, MuLV proviral load and viral expression were significantly attenuated in a dose-dependent manner compared to vehicle controls in BETi-treated, MuLV-exposed cell cultures [15,16,17]. The combined data support conserved molecular interactions and facilitatory effects of BET proteins and gammaretroviral integrases during viral infection.

There is currently no effective treatment for progressive FeLV infection; these cats ultimately develop FeLV-associated terminal illnesses and die within a few years [39], limiting management to supportive therapies and palliative nursing care. This is despite numerous investigations utilizing a variety of drug classes (e.g., nucleoside/nucleotide analogue reverse transcriptase inhibitors, non-nucleoside reverse transcriptase inhibitors, ribavirin (reviewed in [40]), nucleotide synthesis inhibitors [41], integrase inhibitors [11,42], and biomolecular approaches such as interferons [43,44,45], siRNA [46], synthetic peptides [47], and CRISPR-Cas9 [48]. Limitations due to study design, partial efficacy, adverse effects, impracticality for long-term use, and cost have hampered efforts thus far.

Novel therapies promoting conversion from progressive to regressive FeLV infection have the potential to improve long-term prognoses. For example, the interaction of BET proteins with FeLV could be exploited to develop treatment(s) for FeLV infection; considerations include BETi inhibition of the bromodomains of BET proteins or disrupting integrase binding to BET protein ET domains using small molecule ligands or small peptide-based inhibitors as described for MuLV [19,49]. Indeed, reduced in vitro infection and viral spread, as well as delayed tumor activation in a MYC/Runx2 mouse model using replication-competent recombinant MuLV lacking a functional (i.e., BET protein-binding) ET domain-binding motif, have been reported [50].

Several other viruses are reported to interact with BET proteins directly or indirectly as part of their replicative cycles. These interactions and effects are frequently virus-type-specific and may include involvement in viral transcription, the organization of viral replication, host transcriptome modulation, and long-term viral genome maintenance. Effects of BETi can vary from reactivation of latent herpesviruses, cytomegaloviruses, and lentiviruses to overall inhibition of infection (reviewed in [51,52]). For example, inhibition of BRD4 reportedly activates the cGAS/STING/TBK1/IRF3 innate immune pathway, which in turn inhibits the viral attachment of a range of RNA and DNA viruses. Moreover, administration of BETi was protective against pseudorabies virus and vesicular stomatitis virus infection in a mouse model [53]. Thus, BETi can have inhibitory and/or facilitative effects on viral infection depending on the type of virus, highlighting the need for thoughtful consideration and further research.

Archetypal pan-BETi (+)-JQ1 is well characterized and possesses favorable physical characteristics (e.g., dissociation constant, selectivity for BET protein bromodomains). However, (+)-JQ1 also has some undesirable pharmacokinetic properties, such as limited oral bioavailability and a short terminal elimination half-life in vivo, that preclude therapeutic use. Human clinical trials evaluating (+)-JQ1-derived and other drug-class BETi for use as monotherapies or part of combinatorial regimens for various disorders, including cardiovascular disease, diabetes, myelofibrosis, and both solid and hematological malignancies, have been completed or are underway [54]. Newer generation BETi include modifications for increased potency, orally bioavailability, and/or greater terminal elimination half-life compared to (+)-JQ1. Some, such as ABBV-744 and GSK778, are optimized for greater selectivity for one of two distinct BET protein bromodomains in an effort to improve therapeutic indices [55,56].

Due to broad FeLV cell tropism, the effects of BETi on FeLV infection in additional cell lines should be assessed. Furthermore, evaluation of more therapeutically feasible BETi, potential synergistic drug interactions, and the relative importance of the first and second BET protein bromodomains would promote a deeper understanding of the development, prevention, and treatment of FeLV infection.

## 5. Conclusions

These studies of the effects of pan-BETi (+)-JQ1 on two phenotypically disparate cell lines suggest a role for BET proteins as faciliatory mediators of FeLV infection by promoting proviral integration. Notably, (+)-JQ1 treatment significantly reduced total FeLV DNA load and viral protein expression in 81C and 3201 cell cultures challenged with FeLV. Proviral load was significantly decreased in 81C cell cultures, while a trend toward decreased proviral load was observed in 3201 cell cultures treated with (+)-JQ1 and challenged with FeLV. Additional in vivo investigations of BETi dose tolerance, pharmacokinetics, and efficacy of newer-generation BETi as FeLV combinatorial or standalone therapies are warranted.

## Figures and Tables

**Figure 1 viruses-15-01853-f001:**
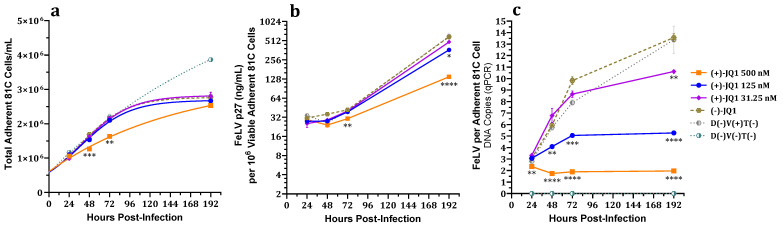
Effects of (+)-JQ1 over time on 81C cell cultures challenged with FeLV. (**a**) No dose-dependent differences in viable adherent cell concentrations were present at 24 and 192 hpi. Dose-dependent decreases in both (**b**) FeLV p27 concentration per million viable adherent cells and (**c**) total FeLV DNA load per adherent cell were observed. Neither FeLV DNA nor p27 were detected in FeLV-unexposed controls. D(−)V(+)T(−) indicates an FeLV-exposed, untreated positive control. D(−)V(−)T(−) indicates FeLV-unexposed, untreated cells. One-way ANOVA with Holm-Šídák multiple comparisons test. Error bars represent ± SEM; *n* = 3 biological replicates. * *p* < 0.05, ** *p* < 0.01; *** *p* < 0.001; **** *p* < 0.0001.

**Figure 2 viruses-15-01853-f002:**
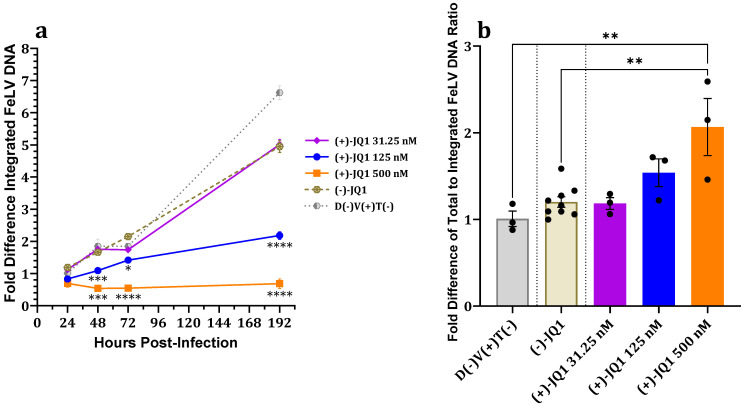
Effects of (+)-JQ1 over time on proviral integration in 81C cell cultures challenged with FeLV. Dose-dependent decreases in fold-differences of (**a**) proviral FeLV DNA load and (**b**) ratio of total-to-integrated FeLV DNA genomes at 24 hpi, suggestive of inhibited proviral integration, were observed. Results were calibrated to D(−)V(+)T(−) at 24 hpi. D(−)V(+)T(−) indicates FeLV-exposed, untreated positive control. One-way ANOVA with Holm-Šídák multiple comparisons test. Error bars represent ± SEM; *n* = 3 biological replicates. * *p* < 0.05; ** *p* < 0.01; *** *p* < 0.001; **** *p* < 0.0001.

**Figure 3 viruses-15-01853-f003:**
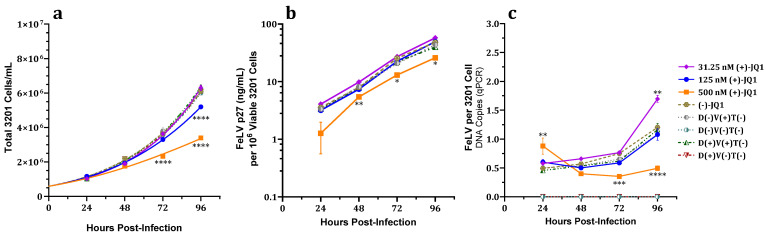
Effects of (+)-JQ1 over time on 3201 cell cultures challenged with FeLV. Dose-dependent decreases in (**a**) total cell concentration, (**b**) FeLV p27 concentration per million viable cells, and (**c**) total FeLV DNA load per cell were observed. No significant changes in precent cell viability were observed (Appendix A). D(−)V(+)T(−) indicates FeLV-exposed, untreated cells. Neither FeLV p27 antigen nor DNA were detected in FeLV-unexposed controls. D(−)V(−)T(−) indicates FeLV-unexposed, untreated cells. D(+)V(+)T(−) indicates FeLV-exposed, DMSO vehicle-treated cells. D(+)V(−)T(−) indicates FeLV-unexposed, DMSO vehicle-treated cells. One-way ANOVA with Holm-Šídák multiple comparisons test. Error bars represent ± SEM; *n* = 3 biological replicates. * *p* < 0.05; ** *p* < 0.01; *** *p* < 0.001; **** *p* < 0.0001.

**Figure 4 viruses-15-01853-f004:**
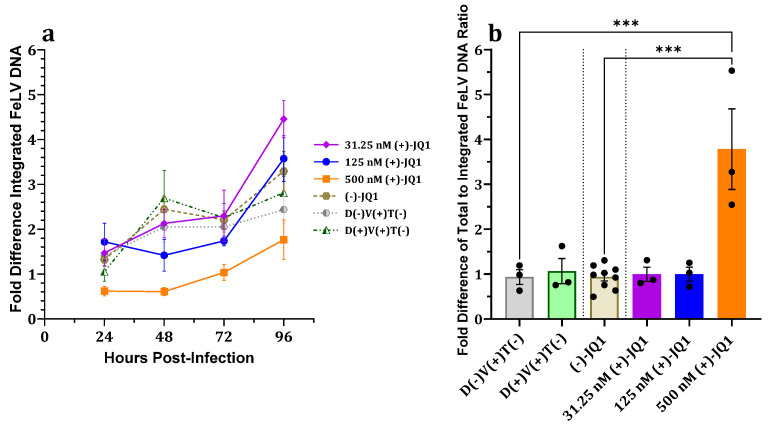
Effects of (+)-JQ1 concentrations over time on proviral integration in 3201 cell cultures challenged with FeLV. A trend toward a fold-difference decrease in (**a**) proviral FeLV DNA load with (**b**) a significant increase in the ratio of total-to-integrated FeLV DNA genomes at 24 hpi was seen. Results were calibrated to D(+)V(+)T(−) at 24 hpi. D(−)V(+)T(−) indicates FeLV-exposed, untreated cells. D(−)V(−)T(−) indicates FeLV-unexposed, untreated cells. D(+)V(+)T(−) indicates FeLV-exposed, DMSO vehicle-treated cells. D(+)V(−)T(−) indicates FeLV-unexposed, DMSO vehicle-treated cells. One-way ANOVA with Holm-Šídák multiple comparisons test. Error bars represent ± SEM; *n* = 3 biological replicates.; *** *p* < 0.001.

## Data Availability

Data presented in this study are available upon request from the corresponding authors.

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
