# Peer review of "BET Inhibitor JQ1 Attenuates Feline Leukemia Virus DNA, Provirus, and Antigen Production in Domestic Cat Cell Lines"

_viruses, 2023, doi:10.3390/v15091853_

Round 1

Reviewer 1 Report (Previous Reviewer 3)

I appreciate the effort that the authors have taken to revise the manuscript. Compared the earlier version, substantial ameliorations are provided which have improved the quality of manuscript. Overall, all my major concerns have been addressed with additional results or explanation.

As a basic study, the authors’ data indicate the promising application of JQ1 as a potential therapy for FeLV infection, which could inspire the future clinical studies. I think this revised manuscript can be considered for publication in Viruses.

Author Response

Reviewer 2 Report (New Reviewer)

In the manuscript entitled “BET inhibitor JQ1 attenuates feline leukemia virus DNA, provirus, and antigen production in domestic cat cell lines,” the authors build off of previous work by other groups and present a thorough workup of BETi, with respect to feline leukemia virus infection. Briefly, the core study is designed around examining FeLV unintegrated DNA, proviral load, and antigen production following variable treatments of the active BET inhibitor and its inactive enantiomer. In concordance with the work performed in MuLV, BETi appears to be a feasible therapeutic target to inhibit proviral integration. The manuscript is thorough, relatively clear, and complete. Publication of these results will help further research in the field. As such, I would like recommend acceptance with minor changes listed below. These suggested changes are an attempt to provide targets for improved concision and clarity:

Author affilition: #3 “Dept of Small Animal Medicine, College of Veterinary Medicine” is incomplete.

Abstract: The phrasing of “(BET) proteins exert a faciliatory effect on MULV…” is unnecessarily confusing. Would it be accurate to just say “(BET) proteins facilitate MuLV replication through proviral integration.”

Line 48: “in contrast” should be “by contrast” when used at the beginning of the sentence

Lines 74-77: “by exerting a dual faciliatory effect” can also just be simplified to X “facilitates” Y and Z.

Lines 138, 155: For the purposes of reproducibility, can you report viral MOI that is represented by 150 and 571.5 uL FL74 viral stock, respectively?

Line 220: There are inconsistencies in the reporting of total FeLV DNA load. As the equation states, the FeLV DNA load is normalized to fALB copy number. However, in many places including line 301 and the figure legends, this calculation appears to be interpreted as copies per cell. As fALB is a single copy haploid gene, the denominator needs to be doubled in order to accurately represent copies per cell.

Line 275: Were any cytopathic effects appreciated in cells, particularly in the 500 nM treatment?

Lines 305 – 307: This statement appears a little out of place and perhaps belongs in the discussion. While you are normalizing p27 antigen against viable cells, I do not see a similar approach for FeLV DNA load. As stated, it would appear as if the normalization by the methods carried out would account for both viable and non-viable cells. Please clarify.

Figures: In general, the figures are busy to look at, difficult to interpret, and do not stand alone without the text. The figure legends do little to explain or walk through the findings and generally appear to restate the Y-axes. Please elaborate and walk through the figure and findings. Furthermore, there are inconsistencies with naming of Y-axes. Figure 1 lists the cell line in A and B, whereas the rest of the axes just say “cells.”

Figures: A suggestion for addressing the business of the figures follows. As the enantiomer treatments are run as control treatments with little variation, it may be easier to display them as one data point. The text can explain that there is no difference between the three treatments. The individual treatments, if necessary, could be reported in supplementary data.

Figure 3A: The presence of the cell viability data crowds the graph. Please move to a separate or supplemental figure.

While the manuscript is written in standard English, there are few sections that are unnecessarily convoluted. Suggestions for changes have been provided in the suggestions for the authors. 

Author Response

This manuscript is a resubmission of an earlier submission. The following is a list of the peer review reports and author responses from that submission.

Round 1

Reviewer 1 Report

Though it seems to be still is a long way to produce an effective drug to treat FeLV-induced disease, such studies are needed, and should be published, as seen e.g. from feline infectious peritonitis that is now curable.

Author Response

Reviewer 1

Primary Comment(s) – Though it seems to be still is a long way to produce an effective drug to treat FeLV-induced disease, such studies are needed, and should be published, as seen e.g. from feline infectious peritonitis that is now curable.

Response to Reviewer 1 Primary Comment(s) – Thank you for your insightful review and comments. We agree that our data will provide greater understanding of FeLV beneficial for the veterinary medical field and feline health.

Reviewer 2 Report

The paper by Garrick M. Moll et al., present the inhibiting of FeLV provirus integration and viral antigen production by the BET inhibitor JQ1 in two feline cell lines. The paper could be categorized as a Brief Report or Communication based on this aspect. However, the mechanism and importance of BET proteins for gamma-retrovirus integration are well-established. Therefore, the in vitro data presented here are as expected. The results seem to lack significance and novelty.

Minor

1. Describe the result shown in Figure 3c, which states that ''significant increases in FeLV DNA load were seen at 24 hpi for 500 nM (+)-JQ1 and at 96 hpi for 31.25 nM (+)-JQ1'' in line 329.  This result was different from others, and it was opposite to the intended purpose.

2. In the 3201 cells experiment, DMSO was used to induce effect on viral infection; However, the data related to DMSO is not described clearly.

3. The authors used (+)-JO1 and sometimes BETi(+)-JQ1, so I would suggest using the same term consistently.

4. How did the authors obtain the cell lines? 

5. I don't think it is necessary to describe future research in the discussion.

Author Response

Reviewer 2

Primary Comment(s) – The paper by Garrick M. Moll et al., present the inhibiting of FeLV provirus integration and viral antigen production by the BET inhibitor JQ1 in two feline cell lines. The paper could be categorized as a Brief Report or Communication based on this aspect. However, the mechanism and importance of BET proteins for gamma-retrovirus integration are well-established. Therefore, the in vitro data presented here are as expected. The results seem to lack significance and novelty. We did anticipate effects, agree it hits expectations, though we believe that not done before in FeLV, a contribution.

Response to Reviewer 2 Primary Comment(s) – We are appreciative of your thorough review and agree that the mechanism and importance of BET proteins for murine leukemia virus (MuLV) integration are well-established. We also note that in vitro molecular data supporting interaction between FeLV integrase and BRD2 has been reported (Gupta et al.). However, there remains a paucity of in vitro cellular data demonstrating that the mechanism and importance of BET proteins for infection extends to other gammaretroviruses, including FeLV. While we agree our data meets expectations, it is nonetheless crucial to demonstrate and verify these points, particularly for the field of veterinary medicine and feline health. Experimental data confirming the mechanisms of disease and therapeutic efficacy are necessary and should be published as part of advancements toward development of potential treatments for FeLV, as seen with coronavirus-induced feline infectious peritonitis that has recently become curable.

Minor Point – Describe the result shown in Figure 3c, which states that ''significant increases in FeLV DNA load were seen at 24 hpi for 500 nM (+)-JQ1 and at 96 hpi for 31.25 nM (+)-JQ1'' in line 329. This result was different from others, and it was opposite to the intended purpose.

Response – “Total FeLV DNA load was increased at the earliest measured timepoint (24 hpi) in 500 nM and at 96 hpi in 31.25 nM (+)- JQ1 treated 3201 cell cultures challenged with FeLV; this may be related to a more rapid replication rate of 3201 versus 81C cells, an increased proportion of unintegrated versus integrated FeLV, or experimental variability.” (lines 378-383)  

Minor Point – In the 3201 cells experiment, DMSO was used to induce effect on viral infection; however, the data related to DMSO is not described clearly.

Response – We have included description of observations for DMSO effects (lines 318-320), which is brief as no apparent differences were observed at the used 0.005% (v/v) DMSO. “No significant differences in viral infection were observed in two additional control groups, FeLV-exposed and FeLV-unexposed 3201 cells with DMSO vehicle alone at the concentration utilized in our experiments [36-38].”

Minor Point – How did the authors obtain the cell lines?

Response – “Cell lines were generously provided by Dr. Lawrence E. Mathes, at The Ohio State University.” (lines 110-111)

Reviewer 3 Report

The authors in this manuscript evaluated the antiviral efficiency of BET inhibitor JQ1 on FeLV replication, and data indicated that JQ1 treatment in two feline cell lines exerts strong suppression on FeLV p27 release, total DNA load as well as proviral DNA level, without causing obvious cell death at working concentrations. The consistent results obtained from two cell lines suggested the potential application of BETi as an antiviral for treatment of FeLV infection and related diseases in cats. The authors have demonstrated that JQ1 curbs viral replication at integration step due to the reduction of proviral DNA after treatment, however, based on current data, it was not convincing to conclude that since earlier steps of FeLV life cycle such as fusion, capsid uncoating, RT and nucleus import. More and sufficient experiments should be carefully conducted targeting the integration and pre-integration steps. Gupta et al reported the interaction between BRD2 and FeLV integrase (ref 16), which could be supporting evidence and the authors should discuss in manuscript.

Other major and minor points are listed below for the authors’ consideration.

Major:

1 Please indicate the MOI of the virus you used for infection. Information from Materials and Methods section showed that you used specific volume (v/v) for infection, which was not scientifically correct nor accurate for other researchers to follow. In addition, if different batches of virus were used, volume (v/v)-based infection may have different viral input and results.

2 In Fig 1a, please explain why JQ1 with 500nM suppressed cell growth at first but showed no effect on cell density at 192h? It seems like at 192h, all cells with different treatment grew into 100% confluent monolayer.

3 Why didn’t set a DMSO control, D(+)V(+)T(-) or D(+)V(-)T(-), in Fig 1c (also Fig 2)? Technically, drug-treated groups should be compared with DMSO addition, not reagent-free, blank condition. Since you also mentioned that DMSO may take potential effect on viral infection (lines 305-306), why only set DMSO control for 3201 cells?

4 In lines 329-330, it would be great if you can explain the reason why 500 nM JQ1 at 24h and 31.25 nM JQ1 at 96h increased the viral DNA? If multiple independent experiments resulted in the same trend, it should not be just an experimental error.

Minor:

1 In line 73, “Figure S1&S2” should be replaced with “Table S1&S2”, which matches the supplementary files.

2 The interpretation of lines 284-285 was confusing and somewhat misleading, which could be “…exceeded drug-induced reductions in 81C cell proliferation”. In comparison, the statement in lines332-333 was much clearer.

3 In line 309, “96 hpi for 125 nM (+)-JQ1” should be “96 hpi for 125 nM and 500 nM (+)-JQ1”.

Author Response

Reviewer 3

Primary Comment(s): The authors have demonstrated that JQ1 curbs viral replication at integration step due to the reduction of proviral DNA after treatment, however, based on current data, it was not convincing to conclude that since earlier steps of FeLV life cycle such as fusion, capsid uncoating, RT and nucleus import. More and sufficient experiments should be carefully conducted targeting the integration and pre-integration steps. Gupta et al. reported the interaction between BRD2 and FeLV integrase (ref 16), which could be supporting evidence and the authors should discuss in manuscript.

Response to Primary Comment(s) – Thank you for your insightful review and comments. It has been shown that gammaretroviruses, including FeLV, do not undergo nuclear import; rather, access of pre-integration complexes to host chromatin only occurs during metaphasic absence of the cell nuclear membrane. “Gammaretroviral DNA gains access to integrate into host cell chromatin and become proviral DNA during nuclear membrane dissolution in the metaphase stage of cell division. Therefore, we measured the effects of (+)-JQ1 on viable cell numbers.” (lines 262-264) A lack of effect of JQ1 on FeLV cellular entry, capsid uncoating, and reverse transcription is supported by minimal variation of total FeLV DNA load (with the exception of 500 nM (+)-JQ1 treated 3201 cells due to their rapid replication rate) amongst the treatment groups during the initial stage of infection (24 hours post-infection), a timepoint by when these viral replication steps should have occurred. These data suggest that reduction of viral replication by JQ1 was not related to these early cell cycle steps.

Major:

Comment 1 – Please indicate the MOI of the virus you used for infection. Information from Materials and Methods section showed that you used specific volume (v/v) for infection, which was not scientifically correct nor accurate for other researchers to follow. In addition, if different batches of virus were used, volume (v/v)-based infection may have different viral input and results.

Response 1 – We appreciate you raising this significant challenging point. Identification of foci is subjective and can be poorly reproducible between individuals. FeLV p27 ELISA (equivalent of HIV-1 p24 ELISA) and measurement of reverse transcriptase units were considered, but not utilized because they do not necessarily reflect infectious titer. Therefore, we used a standard operating procedure to create our viral stocks: initial passaged FL-74 cell concentration of 2.1 × 106 viable cells/mL ± 5% in T175 flasks harvested 40-48 hours after passage (lines 126-130). Culture medium formulation is provided (lines 103-108). Our intra-experiment comparisons are valid because the same viral stock was utilized for all virus-exposed cell wells within an experiment. No comparisons between 81C and 3201 cell FeLV infectivity and effects of (+)-JQ1 are made. Furthermore, inter-experimental results and interpretations were repeatable for each cell line with different batches of virus (data not shown).

Comment 2 – In Fig 1a, please explain why JQ1 with 500nM suppressed cell growth at first but showed no effect on cell density at 192h? It seems like at 192h, all cells with different treatment grew into 100% confluent monolayer.

Response 2 – Viability of adherent 81C cells was not affected among all groups, though adherent cell count doubling time was longer for 500 nM (+)-JQ1 at 48 and 72 hpi. Other groups reached 100% confluency before the 500 nM (+)-JQ1 treatment group; by 192 hours 500 nM (+)-JQ1 group reached 100% confluency as well.

Comment 3 – Why didn’t set a DMSO control, D(+)V(+)T(-) or D(+)V(-)T(-), in Fig 1c (also Fig 2)? Technically, drug-treated groups should be compared with DMSO addition, not reagent-free, blank condition. Since you also mentioned that DMSO may take potential effect on viral infection (lines 305-306/), why only set DMSO control for 3201 cells?

Response 3 – The biologically inactive enantiomer (-)-JQ1 groups contained the same concentration DMSO vehicle as the (+)-JQ1 groups, thereby serving as fundamentally superior version of vehicle control for all experiments. Though unnecessary, the D(+)V(+)T(-) group was added to 3201 cell  experiments as a comprehensive comparison. “No significant differences in viral infection were observed in two additional control groups, FeLV-exposed and FeLV-unexposed 3201 cells with DMSO vehicle alone at the concentration utilized in our experiments [36-38].” (lines 318-320)

Comment 4 – In lines 329-330, it would be great if you can explain the reason why 500 nM JQ1 at 24h and 31.25 nM JQ1 at 96h increased the viral DNA? If multiple independent experiments resulted in the same trend, it should not be just an experimental error.

Response 4 – “Total FeLV DNA load was increased at the earliest measured timepoint (24 hpi) in 500 nM and at 96 hpi in 31.25 nM (+)- JQ1 treated 3201 cell cultures challenged with FeLV; this may be related to a more rapid replication rate of 3201 versus 81C cells, an increased proportion of unintegrated versus integrated FeLV, or experimental variability.” (lines 378-383